# Green Extraction at Scale: Hydrodynamic Cavitation for Bioactive Recovery and Protein Functionalization—A Narrative Review

**DOI:** 10.3390/molecules31010192

**Published:** 2026-01-05

**Authors:** Francesco Meneguzzo, Federica Zabini, Lorenzo Albanese

**Affiliations:** Institute of Bioeconomy, National Research Council of Italy, Via Madonna del Piano 10, 50019 Firenze, Italy; federica.zabini@cnr.it (F.Z.); lorenzo.albanese@cnr.it (L.A.)

**Keywords:** hydrodynamic cavitation, green extraction, pectin–polyphenol conjugates, dry protein isolate, protein–polyphenol conjugates, bioavailability, circular economy, nutraceuticals

## Abstract

Hydrodynamic cavitation (HC) is a green and readily scalable platform for the recovery and upgrading of bioactives from agri-food and forestry byproducts. This expert-led narrative review examines HC processing of citrus and pomegranate peels, softwoods, and plant protein systems, emphasizing process performance, ingredient functionality, and realistic routes to market, and contrasts HC with other green extraction technologies. Pilot-scale evidence repeatedly supports water-only operation with high solids and short residence times; in most practical deployments, energy demand is dominated by downstream water removal rather than the extraction step itself, which favors low water-to-biomass ratios. A distinctive outcome of HC is the spontaneous formation of stable pectin–flavonoid–terpene phytocomplexes with improved apparent solubility and bioaccessibility, and early studies indicate that HC may also facilitate protein–polyphenol complexation while lowering anti-nutritional factors. Two translational pathways appear near term: (i) blending HC-derived dry extracts with commercial dry protein isolates to deliver measurable functional benefits at low inclusion levels and (ii) HC-based extraction of plant proteins to obtain digestion-friendly isolates and conjugate-ready ingredients. Priority gaps include harmonized reporting of specific energy consumption and operating metrics, explicit solvent/byproduct mass balances, matched-scale benchmarking against subcritical water extraction and pulsed electric field, and evidence from continuous multi-ton operation. Overall, HC is a strong candidate unit operation for circular biorefineries, provided that energy accounting, quality retention, and regulatory documentation are handled rigorously.

## 1. Introduction

The valorization of agri-food and forestry byproducts represents a cornerstone of the circular bioeconomy, with global waste streams exceeding 1.3 billion tons annually [1]. The fruit processing industry contributes substantially, with citrus processing alone generating about 15 million tons of peel waste (CPW) [2] and pomegranate processing more than 1.5 million tons [3]. On the other hand, forestry residues account for 60% of harvested biomass [4]. These underutilized resources harbor valuable bioactive compounds—flavonoids, pectin, and terpenes—with proven health benefits ranging from cardioprotection [5,6] to hepatoprotection and neuroprotection [7,8,9].

Conventional extraction methods (e.g., Soxhlet and maceration) remain widely used but are often constrained by long processing times, large solvent volumes, multistep separations, and limited suitability for industrially relevant throughput [10]. Technologically, these methods suffer from low yields (typically <30% for polyphenols) [11], extended processing times (up to 24 h) [12], and poor scalability to industrially relevant volumes [13]. Economically, solvent costs and purification steps consume 40–60% of operational expenses [14], while regulatory hurdles like the EU Novel Food Regulation (2015/2283) impose lengthy approval processes for extracts obtained via novel technologies [15].

Several green intensification techniques, including ultrasound-assisted extraction (UAE), microwave-assisted extraction (MAE), and pulsed electric field (PEF), can improve extraction kinetics and reduce the solvent burden, but each faces scale constraints. For example, acoustic field non-uniformity can hinder UAE beyond pilot volumes (reactors > 50 L) [16]. MAE requires thermal management to avoid degradation at power levels roughly >5 kW [17], and PEF is most often deployed as a pre-treatment step that still requires downstream extraction [18].

Hydrodynamic cavitation (HC) differs from these approaches by delivering high energy density directly to particles in a flowing liquid or slurry while remaining compatible with simple hydraulic scale-up. Besides intensifying physical, chemical, and biochemical processes, HC can promote radical-mediated pathways through hydroxyl radical formation under cavitation conditions [14]. A striking illustration of HC capabilities is the drastic acceleration of the temperature-dependent conversion of S-methyl-methionine to dimethyl sulfide and its rapid degassing in brewer’s wort, which were historical bottlenecks in conventional wort boiling [19].

In natural product extraction, HC has been reported as highly competitive in terms of both extraction yield and process yield (amount of bioactives recovered per unit energy consumed) and—particularly for linear static reactors such as Venturi devices—as a straightforwardly scalable technology [14]. Beyond efficiency, HC has repeatedly shown the ability form stable phytocomplexes, including conjugates of low-methoxy pectin with flavonoids and volatile terpenoids in citrus matrices [20], likely (at least with flavonoids) through both non-covalent and stronger covalent bonds, the latter enabled by the excess generation of hydroxyl radicals [8]. Such assemblies, obtained without additional formulation steps, support controlled release and sustained biological activities (e.g., antioxidant, antimicrobial, and anti-inflammatory) [20] and in vivo effectivity at lower doses than isolated compounds [6,8]. Similar observations in pomegranate byproducts raise the possibility of analogous conjugation phenomena also in high-methoxy pectin systems [21].

HC showed more than sevenfold higher efficiency compared to the hot water technique with the extraction of spruce bark as a byproduct of conifer wood supply chains [4] while retaining volatiles which were deemed responsible for the higher antibacterial activity.

Given the potential for HC to enable the sustainable exploitation of vegetable byproducts from agri-food and forestry supply chains, nutraceuticals and food supplements appear as an obvious outlet, also considering their astonishing revenues surpassing USD 180 billion, with projections for further sustained growth [22]. However, the sector is crowded, and new products must compete in a market with increasing scrutiny on safety, quality, and claim substantiation [22,23].

Here, we therefore consider a pragmatic commercialization route that may be smoother than launching stand-alone supplements: integrating HC-derived bioactive extracts into dry protein isolate (DPI) products, which are already widely consumed and growing in demand. Whey protein isolate (WPI), a highly valuable and easily digestible source of amino acids, represents a large and expanding market, amounting to nearly USD 9 billion in 2023 and projected to more than double in 2030 [24], while plant-based DPIs continue to gain share driven by dietary preferences and environmental concerns [25].

Across all potential sources of bioactive extracts suitable for the functionalization of DPI products, this study focuses on the byproducts of the supply and processing chains of two classes of natural products, citrus fruits and conifer plants. Such sources are quite familiar to the authors, sufficiently different in nature and composition and increasingly considered for valorization through extraction due to their recognized biological activities and the widespread availability of the raw materials. Citrus byproducts are an agri-food resource dominated by polysaccharide compounds, while conifer plant byproducts are a forest resource dominated by lignocellulosic material.

Beyond the direct blending of DPI products with dry polyphenol-rich extracts, this study briefly reviews the emerging distinct features of the HC-based extraction of proteins from plant sources and the prospective opportunity and potential advantages of coprocessing DPI products with the HC-derived isolated polyphenols of polyphenol-rich phytocomplexes.

## 2. Selection Method of HC-Related Literature

This is a narrative, expert-judgement review rather than a systematic review. Study selection was purposive: papers were included when, in the authors’ assessment, they materially informed process understanding, energy/use-of-water implications, product functionality, or scale-up of HC methods within agri-food/forestry streams. Therefore, the coverage is illustrative and may omit relevant studies. The intent was to synthesize practice-relevant evidence, surface consistent patterns and constraints, and delineate priorities for future systematic evaluations.

To increase transparency and reduce selection bias, we complemented expert curation with a structured Scopus search conducted on 22 December 2025 over the period 2000–2025 using the following query: TITLE-ABS-KEY (“hydrodynamic cavitation” AND (extract* OR valorization OR polyphenol* OR pectin OR flavonoid* OR tannin* OR protein*)) AND NOT TITLE-ABS-KEY (wastewater OR “advanced oxidation” OR desulfur* OR “microbubble”). The query returned 249 records, 63 of which (25.3%) were also indexed in PubMed, as an approximate indicator of health-relevant coverage. Appendix A provides the raw export, including the fields of authors, title, publication year, doi, PubMed ID, and Scopus link. Appendix A shows the annual publication trend, while Appendix A shows the most frequent countries in affiliations (top 20). Microsoft^®^ Excel^®^ for Microsoft 365 MSO (Version 2509, Microsoft, Redmond, WA, USA) was used to organize the data and produce the charts shown in Appendix A, as well as the charts shown in Appendix A.

Based on Appendix A, the publication trend started increasing after the year 2017, with a remarkable increase in 2024 and an even greater jump in 2025, apparently pointing to a growing interest in the HC-based processing of natural products. Based on Appendix A, the top 20 countries in affiliations were led by Italy, followed by India, China, and Ireland.

## 3. Comparative Analysis of Green Extraction Techniques of Bioactive Compounds

Among the resources focused on in this study, CPW has been the most studied regarding extraction methods, with sufficient literature about both conventional and novel green methods, and will be the focus of this section. As the widely adopted green extraction principles (GEPs), recently formulated within a rigorous sustainability benchmark by de Souza Mesquita and coworkers [26], foresee the use of water as the only solvent or at least safe solvents (principle 1), non-denaturing conditions, implying mild extraction environments (principle 2), and the possibility to use renewable energy sources (principle 5), only novel green methods are discussed in the following. Among these methods, to the best knowledge of the authors, only subcritical water extraction (SWE) and HC have also been tested on conifer byproducts.

Similarly to Section 2, study selection for novel green extraction methods other than HC was subjective and based on the authors’ expertise, broadly covering the most recent papers enabling a meaningful comparison with HC.

### 3.1. Ultrasound-Assisted Extraction

UAE methods are the most common across non-conventional green methods to obtain phenolic and volatile compounds from citrus byproducts. UAE is a solid–liquid extraction method, based on the phenomenon of acoustic cavitation, i.e., generating vapor bubbles within a liquid, the implosion of which causes the fragmentation of molecule cell walls, increases the exchange surface, and promotes the release of bioactive compounds [27].

Cavitation in liquid media is a multiphase phenomenon consisting of the generation, growth, and quasi-adiabatic collapse of vapor-filled bubbles under an oscillating pressure field, resulting in pressure shockwaves (up to 1000 bar), hydraulic jets, extreme local temperatures (up to thousands of K), and the generation of free radicals [28,29].

The main parameters affecting the recovery efficiency are particle size, temperature, time, solvent type, and ultrasonic power and frequency [16]. Using water as the only solvent, the extraction rate of phenolic acids (*p*-coumaric, caffeic, and chlorogenic) and hesperidin from *Citrus limon* L. pomace (peel, membranes, and seeds) significantly increased with increasing temperature and decreasing particle size [30]. In the case of pectin recovery, the extraction rate negatively correlated with the liquid-to-solid ratio, with a significant decrease in pectin yield with increasing solvent volume [31]. Several studies were performed to validate and optimize UAE for the extraction of polyphenols, pectin, essential oils, and carotenoids [32], although transition to full scale appears intrinsically problematic [13].

### 3.2. Pulsed Electric Field

PEF is an emerging non-thermal method for the extraction of bioactive compounds, based on the application of microsecond pulses of high-intensity electric fields that induce a non-reversible electroporation in cell membranes, with the cell disintegration index used to evaluate the appropriate PEF treatment conditions [18]. The intensity of the electric field, affecting the structural characteristics of the target material, is the most important determinant of the extraction yield, followed by the frequency and amplitude of the pulses, the wave shape, and the exposure time [33]. The temperature of the solvent in the treatment chamber is a decisive parameter for the diffusion of electrical pulses, with the process temperatures of extractive processes usually below 90 °C, both to ensure the preservation of functional compounds and to avoid the reduction in water viscosity that occurs at high temperatures.

PEF is also a technique applied in the pre-treatment of biomass to increase the extraction rate in subsequent processes. Hwang et al. compared the conventional flavonoid extraction of *Citrus unshiu* with PEF followed by SWE, with an increase in the hesperidin concentration by about 22.4, 2.1, and 1.2 times in comparison with hot water, methanol, and SWE, respectively [34]. Moreover, Luengo et al. found an increase of 20 to 159% in the extraction yield of the most abundant polyphenols from orange peels, like naringin and hesperidin, applying PEF before conventional pressing [35].

As a pre-treatment for orange peels, PEF also showed advantages in increasing the EO extraction yield, with a 33% increase in limonene extraction when combined with ethanol [36]. Notably, the selectivity of the extracted aromatic compounds depended on the type of solvent used and the relative affinity to intracellular compounds that solubilize easily following treatment with PEF.

### 3.3. Microwave-Assisted Extraction

MAE takes advantage of the dielectric properties of the plant matrix, sustaining the interaction of the polar molecules of the sample with the solvent, which in turn causes rapid heating and rupture of plant cell walls. MAE operates through two mechanisms of heat generation: dipolar rotation, with water evaporation, and ionic conduction through the resistance of the solution to ion flow, producing friction and heat [27].

Combined MAE with solvent (ethanol) extraction showed superior flavonoid recovery from orange peels compared with conventional methods, operating at low temperatures and over a short time [37]. However, optimization of the microwave power and the exposure time to microwaves is critical to avoid degradation of phenolic and thermolabile compounds [17]. MAE, combined with steam diffusion, was used to increase the extraction yield of EOs from citrus byproducts, such as D-limonene, reducing the processing time by about 3 to 20 times and achieving a better score on the aroma sensory profile [38].

### 3.4. Enzyme-Assisted Extraction

Enzyme-assisted extraction (EAE) can be useful in the case of particularly tight molecular bonds that are difficult to cleave. The enzyme used hydrolyzes the cell walls, releasing the target compounds contained in the plant biomass [39]. The main parameters that affect the extraction yield of phenolics while preserving their biological properties are the condition of the peels, the temperature of the extraction, the pH, the types of enzymes, the enzyme concentration, and the citrus species [40]. EAE offers several advantages in pectin extraction from plant resources such as CPW because this technique can increase the pectin extraction yield compared to traditional processes and can be used under low process temperatures, thereby reducing the energy consumption [41]. EAE also acts as an effective CPW pre-treatment method to improve the distillation of EOs, with an increase in extraction yields by up to 6-fold compared to traditional steam distillation, and it presents an opportunity for the complete reuse of plant biomass, as it generates high amounts of sugars following enzymatic hydrolysis [42]. Finally, EAE can be coupled with UAE, favoring the extraction of compounds most closely bound to the raw material, replacing heat pre-treatment, reducing the ultrasonic extraction time, and consequently increasing the yield by as much as 2 times [43].

### 3.5. Subcritical Water Extraction

The SWE method involves the use of water at temperature and pressure conditions below the critical point of 374.15 °C and 22.1 MPa. The change in chemical and physical properties allows the selective extraction of polar and non-polar compounds [44] and the extraction of hydrophilic and lipophilic products [45]. SWE was used for the enhancement of the extraction yield of mandarin (*Citrus unshiu* Markovich) peels at laboratory (1 g of sample) and pilot scales (100 g of sample), with a processing time of between 10 and 15 min, finding similar yields of total phenolic compounds and an increase in the extraction yield of hesperidin and narirutin at 130 °C and a subsequent decrease at 150 °C, while naringenin showed higher yields at 170 °C [46].

Compared to conventional organic solvent extraction, SWE showed a higher extraction yield for hesperidin (1.4 to 5.8 times), narirutin (1.1 to 5.6 times), and polymethoxyflavones (from 1.1 to 1.6 times), with the extraction yield of citrus flavonoids directly correlating to the temperature and flow rate [47]. These results are consistent with other studies concerning the extraction yield and antioxidant activity of flavanones derived from defatted orange peels [48]. SWE was also used in two-step processes for the extraction of citrus flavanones (hesperidin and narirutin) at 150 °C and mono- and disaccharides at 200 °C [49].

### 3.6. Natural Deep Eutectic Solvents

Natural Deep Eutectic Solvents (NADESs) are eutectic solvents that use only primary metabolites as solvents, such as sugars, organic acids and bases, and amino acids [27]. NADESs can also work in a multistep integrated process to extract different compounds (D-limonene, proteins, and polyphenols) with a unique solvent like cholinium chloride or ethylene glycol as a hydrogen bond donor [50]. The effect of NADESs can be enhanced through their combination with UAE, achieving a synergy between mass exchange and the disruption of cell membranes performed by UAE and the stabilization of bioactive compounds provided by NADESs [51]. Among the most promising NADESs for application to CPW are lactic acid–glucose with a yield for the total phenolic content of 1932 ± 7.83 mgGAE/100 gdw and L-proline–malic acid with a yield of 2164 ± 5.17 mgGAE/100 gdw, also showing good stability of polyphenols after 30 days of storage (25 °C and 4 °C) and lower degradability compared to ethanol extracts [52].

### 3.7. Hydrodynamic Cavitation

HC, sharing with UAE the exploitation of the cavitation phenomena in liquid media, is performed either by circulating a liquid through static constrictions of various geometries or by special immersed rotary equipment. Contrary to UAE, HC is a straightforwardly scalable technological solution, showing outstanding effectiveness and efficiency for food processing, process intensification, and extraction of natural products, alongside plenty of other applications [13,53]. Direct pilot-scale experiments, using water as the only solvent, with 42 kg of fresh orange byproducts in 120 L of water, afforded the extraction of a low-methoxy pectin (degree of esterification of 17.05%) rich in adsorbed hesperidin, naringin, other polyphenols, and terpenes.

The application of HC to the extraction of CPW allowed a completely new class of phytocomplexes to be obtained, dubbed IntegroPectin and consisting of stable conjugates of pectin, flavonoids, and volatile compounds (terpenes) [2,54], with remarkable pharmacological activities and much higher bioavailability compared to isolated flavonoids. The role of cavitation in the production of IntegroPectin was later confirmed by experiments using UAE [55].

The phytocomplex effect, where conjugated molecules show greater bioactivity than isolated compounds, was further evidenced by the striking contrast of the in vitro and in vivo anti-inflammatory performances of red orange IntegroPectin [8], which also exhibited a remarkable degree of standardization using batches of raw material collected in different years and at different times of the harvesting season.

HC-based extraction processes applied to CPW from lemon and grapefruit also revealed the stable conjugation of pectin with flavonoids and terpenes [56,57], spontaneously achieving a result comparable to complex manufacturing processes, for example, used to create hydroxytyrosol–pectin conjugates [58]. An HC-based grapefruit extract was used in an in vivo study, showing anti-ischemic cardioprotective activity far exceeding that of the pure bioactive flavanone naringenin on a dose-dependent basis [6]. It is also notable that insoluble residues of HC-based CPW extraction processing mainly consisted of highly micronized cellulose with high technical value [59], thus further contributing to the citrus circular economy.

Reactor/extractor design strongly determines the hydrodynamic conditions experienced by the biomass slurry [14]. In static linear devices (Venturi tubes and orifice plates), the cavitation intensity is primarily controlled by the imposed pressure drop (ΔP), the local velocity at the throat/orifice, and the resulting cavitation number (σ), while bulk temperature also matters through its effect on vapor pressure and cavitation inception. Venturi geometries generally provide partial pressure recovery and reduced clogging, enabling a wider operating window and lower specific energy, whereas orifice plates can yield higher local shear at the expense of higher irreversible losses and potentially stronger erosion/material wear. Rotational devices (e.g., rotor–stator or vortex-based units) generate cavitation through high-shear swirling flow and are often operated at higher power densities; therefore, defensible comparison and scale-up require reporting the characteristic velocity (e.g., tip speed), power draw, residence time, and number of passes. For solids-rich slurries, viscosity and loading feed back into ΔP and σ, so pump/control settings and temperature management (recirculation time, heat exchange) should be explicitly reported to preserve thermolabile compounds and to keep operating conditions consistent across scales. Recent pilot-scale demonstrations underline that maintaining the σ, temperature, and pass number—rather than merely scaling volume—is key to reproducible extraction/functionalization outcomes [60].

### 3.8. Summary of Extraction Techniques

Table 1 summarizes the main advantages and drawbacks of the methods considered for the extraction of bioactive compounds from CPW.

Across the CPW extraction methods, HC is frequently reported as highly competitive for pilot-scale, water-based extraction, particularly where scalability, low solvent use, and process integration are primary constraints [26], although the persistent lack of technological and process standardization represents a challenging task that needs to be urgently addressed [61].

Cross-study comparisons are limited by heterogeneity in biomass, reactor geometry, and reporting endpoints; in particular, performance and greenness remain context-dependent and are best established via head-to-head comparisons using harmonized metrics (e.g., yield/selectivity, specific energy, and downstream separability). Truly head-to-head benchmarking remains rare; notably, Tienaho et al. directly compared pilot-scale HC with SWE on the same spruce bark feedstock, enabling a more defensible method-to-method comparison than cross-study contrasts, highlighting both the HC process yield, which is seven times higher than that of SWE, and a few criticalities, including the particle size limitation still affecting HC [4].

Figure 1 shows a typical HC process flow for green extraction and phytocomplex formation; to achieve standardization, at least the following metrics should be reported, with such elements having been extensively documented in past studies [62]: the cavitation number, pressure drop, specific energy consumption, temperature rise, solids loading, and pass number/residence time distribution.

Table 2 provides a qualitative, evidence-bounded assessment of how selected extraction technologies may align with key green extraction principles (GEPs). The matrix is not intended as a ranking: many criteria are strongly matrix-, solvent-, and scale-dependent, and robust conclusions require standardized reporting and, ideally, direct comparative experiments.

Overall, HC can align well with several GEPs in reported pilot-scale implementations (notably water-based operation and scale-up feasibility). Nevertheless, multiple criteria are inherently context-dependent (e.g., filtration/clarification with high solids, choice of drying route, and energy accounting). Accordingly, we avoid single-number “compliance” scores and instead emphasize transparent reporting and balanced technology selection by target product and constraints.

## 4. Bioactive Compounds: In Vivo, Ex Vivo, and Clinical Evidence

Consumers of DPI products, either animal- or plant-based, are usually quite aware of their benefits for health and sports performance. Thus, the functionalization of DPIs with bioactive compounds, either integral or purified phytocomplexes extracted from natural products, should be based on robust evidence of the additional value, confirmed by clinical trials, in vivo and ex vivo experiments, or computational predictions. Moreover, further ingredients should be as cheap as possible to avoid increasing the price of functionalized DPIs, which points to HC as the most efficient extraction and processing technique also leading to higher bioavailability and consequently lower effective doses.

In the following, recent evidence is presented regarding the biological functions of the dominant bioactive compounds or integral phytocomplexes extracted from resources associated with documented HC-based extraction processing: orange peel [63], pomegranate peel [64], and silver fir (*Abies alba*) byproducts [65].

### 4.1. Orange Peel Extracts

Hesperidin is by far the dominant flavanone in orange peel extracts and stands out for its many biological functions. It shows convergent evidence across clinical, in vivo, and in silico studies, with green extraction routes enabling high-yield recovery (e.g., HC, hydroalcoholic extraction). In a randomized controlled trial, oral supplementation with a citrus-derived hesperidin formulation (with alpha-glucomannan phosphate, soy proteins, and spermidine; food-grade extract, typically hydroalcoholic from peels) improved immune function, lowered biological age, and ameliorated the oxidative–inflammatory status [66]. In vivo, a hesperidin-rich red orange byproduct phytocomplex produced via HC in water (dubbed AL0042) countered thioacetamide-induced minimal hepatic encephalopathy in mice, showing neuroprotective, anti-inflammatory, and antioxidant actions [8]. Complementing safety and the mechanism, an HC-derived citrus peel extract showed low acute toxicity, antioxidant activity, and stimulated immune responses in murine models [67].

Computational predictions reinforce the disease-modifying potential: hesperidin inhibited α-synuclein aggregation in molecular dynamics analyses, supporting anti-amyloidogenic activity relevant to neurodegeneration [68]. Systems-level reviews further integrate ex vivo and clinical dermatology data, highlighting the skin barrier support, anti-UV/anti-inflammatory effects, and vascular protection of peel-derived hesperidin formulations [69]. Finally, docking-led appraisals during the COVID-19 era consistently flagged hesperidin as a top-rank binder of viral/host targets while advocating scalable production via HC [63].

Collectively, orange-peel-derived hesperidin—obtained by water-based HC or conventional hydroalcoholic extraction—exhibits immunomodulatory, antioxidant, anti-inflammatory, neuroprotective, and anti-amyloidogenic functions across clinical, in vivo, and computational lines, with ex vivo/clinical dermatology evidence emerging for skin applications.

### 4.2. Pomegranate Peel Extracts

Punicalagin—the signature ellagitannin of pomegranate peel—anchors a growing body of translational evidence, with scalable recovery by hydroalcoholic or water-based methods, including HC. An HC pomegranate byproduct extract (peel–pomace; water, no added solvents) showed superior cardiovascular actions in rodent models and vascular ex vivo assays [21].

A randomized, double-blind, placebo-controlled clinical trial of an oral pomegranate extract (standardized peel/fruit polyphenols from solvent extraction) improved skin wrinkles and biophysical features while modulating the gut–skin axis [70]. In patients with type 2 diabetes, a standardized pomegranate peel extract (punicalagin-rich; solvent-extracted) improved the plasma lipid profile, the fatty acid balance, and blood pressure [71]. A comprehensive review of clinical studies corroborates the cardiometabolic and anti-inflammatory benefits of pomegranate preparations rich in punicalagin and related ellagitannins [72].

Neuroinflammation models further support efficacy: a new peel extract formulation (punicalagin-dominated ellagitannins; solvent extraction) alleviated disease severity in mice with experimental autoimmune encephalomyelitis [73]. Photoprotection findings with an aged pomegranate extract extend to a clinical setting. A randomized, double-blind, placebo-controlled clinical trial evaluated the impact of an orally administered standardized pomegranate extract on UV-induced erythema, melanin content, skin hydration, and lightness. The extract demonstrated high antioxidant capacity and significantly reduced reactive oxygen species (ROS) and inflammatory cytokine levels in vitro. Clinical findings showed significant reductions in UV-induced erythema and melanin levels, with concurrent improvements in skin hydration and lightness compared to the placebo [74].

Computational predictions converge with these outcomes: docking and absorption, distribution, metabolism, excretion, and toxicity analyses indicate that punicalagin and other peel metabolites hit multiple protein targets with antidiabetic relevance (e.g., α-amylase and α-glucosidase), supporting glucose- and lipid-regulatory activity [75].

Overall, punicalagin-rich pomegranate extracts, also obtained via HC in water or conventional hydroalcoholic extraction, demonstrate clinical, in vivo, and ex vivo benefits spanning vascular protection, dermal photoprotection, neuroinflammation mitigation, and cardiometabolic control, with reviews synthesizing consistent human evidence [76].

### 4.3. Abies alba Extracts

*Abies alba* extracts have been tested in vivo and ex vivo. In vivo, a trunk–wood extract (polar solvent extraction; lignans, phenolic acids, flavonoids) protected guinea pig arteries against atherogenic diet damage; vasoprotective, antioxidant, and anti-inflammatory actions were reported [77]. Ex vivo, an *Abies alba* extract (wood/bark; polyphenols) reduced infarct size and improved post-ischemic recovery in isolated rat hearts (Langendorff model), showing cardioprotective and anti-lipid-peroxidation activity [78]. An ex vivo Thiobarbituric Acid-Reactive Substance (TBARS) assay performed on a homogenized rat brain showed that an HC-based extract of silver fir twigs inhibited lipid peroxidation better than extracts of *Picea abies* twigs or bark [9].

Mechanistic support came from a wood extract enriched in lignans (e.g., matairesinol, pinoresinol), obtained using solvent extraction with fractionation. Robust gastrointestinal stability and systemic antioxidant activity were demonstrated [79]. Broader softwood work that included *Abies alba* bark/needle preparations (hydroalcoholic/water extracts, with phenolics, flavonoids, and terpenes) showed antioxidant and wound-healing bioactivity; in vitro and in vivo assays; and pro-regenerative, antioxidative, and anti-inflammatory functions [80]. In skin-relevant ex vivo cell models, metabolomics on human keratinocytes exposed to softwood knot wood extracts comprising *Abies alba* revealed a shift toward antioxidant and anti-inflammatory phenotypes [81].

Overall, *Abies alba* extracts—obtained mainly via hydroalcoholic or aqueous techniques from bark, trunk/wood, branches, twigs, and needles—feature lignans, proanthocyanidins, phenolic acids, flavonoids, and terpenes as dominant classes and consistently express antioxidant, anti-inflammatory, cardioprotective, vasoprotective, and wound-healing bioactivities across clinical, in vivo, and ex vivo studies.

## 5. Direct Blending of DPIs with HC-Based Bioactive Extracts

HC-based dry extracts, such as spray-dried IntegroPectin, can be physically blended with commercial DPIs. Obvious technical benefits include process simplicity, dose flexibility, and the use of existing dry blenders. Functional benefits can include an increase in the overall antioxidant, anti-inflammatory, or cytoprotective activity of the blended mixture compared to pure DPIs, similar to gluten-free biscuits with 2.5 wt% of the rice flour replaced with lemon IntegroPectin, which also showed unchanged or improved texture and sensory properties compared to untreated biscuits [82], or gluten-free biscuits supplemented with extracts of pomegranate byproducts, which substantially extended the shelf life [83]. Wholewheat bread supplemented with extracts of silver fir needles, as well as spruce, pine, and fir twigs, showed remarkably increased antioxidant activity [84,85].

Assessing the effective amount of further bioactive ingredients, which, when added to DPIs, potentially result in significant biological effects, could help standardization and resource and cost containment. The following calculations were performed assuming the usually recommended serving of commercial whey or plant-based DPIs of about 30 g daily, which agrees with the recommendations of the International Society of Sports Nutrition aimed at maximally stimulating muscle protein synthesis [86]. Moreover, when a clinically relevant dose of a specific bioactive compound or phytocomplex was not available but in vivo trials provided effective doses for animals, dose translation from different animals to humans was performed according to consolidated conversion factors [87], assuming an average human weight of 70 kg.

Based on an in vivo effective dose of an HC-based integral extract of red orange waste peel (IntegroPectin) of 20 mg/kg daily in mice concerning hepatoprotective and neuroprotective effects, equivalent to approximately 200 mg per day for humans (about 5 mg daily of hesperidin), an effective proportion of red orange IntegroPectin to DPI would be 1:150 *w*/*w*, i.e., 1 g of red orange IntegroPectin per 150 g of whey DPI. The in vivo effective dose of an HC-based integral extract of grapefruit waste peel (IntegroPectin) was 135 mg/kg in rats concerning anti-ischemic cardioprotective activity [6], equivalent to approximately 1.5 g daily for humans. This means an effective proportion of grapefruit IntegroPectin to DPI of 1:20 *w*/*w*.

A phytocomplex extracted using HC from pomegranate byproducts showed an in vivo effective dose of 100 mg/kg in rats concerning chronic hypotensive activity, along with anti-inflammatory and anti-fibrotic activities [21], equivalent to approximately 1.1 g daily for humans (about 55 mg of punicalagins). This means an effective proportion of whole-pomegranate extract to DPI of 1:27 *w*/*w*. Clinical doses for pomegranate extracts span from 250 mg/day of standardized peel extract (about 75 mg punicalagins, oral supplementation) for skin outcomes (significant improvements in several biophysical properties and wrinkles and shifts in the skin microbiome) [70] to 250 mg twice a day (about 35 mg total punicalagins, hydroalcoholic extraction, oral supplementation) in type 2 diabetes [71], which translate into an effective proportion of pomegranate extract to DPI of 1:120 and 1:240 w/w, respectively.

Overall, relatively small amounts of extracts of citrus and pomegranate fruit byproducts would be needed to effectively functionalize both animal- and plant-based DPIs (1/20 to less than 1:200 *w*/*w*), depending on the origin and nature of the phytocomplex and the target functionality.

Extracts of forestry byproducts, including *Abies alba* extracts, represent another valuable resource for the functionalization of DPIs through direct blending. In a human meal challenge, one capsule of Belinal^®^ (Alpe Pharma, Ljubljana, Slovenia), containing 200 mg of *Abies alba* wood extract, administered concomitantly with 100 g of white bread, significantly reduced post-prandial glycemia with effects comparable to the positive control acarbose [88]. This means an effective proportion of *Abies alba* extract to DPI of 1:150 w/w. For 8 weeks, guinea pigs were fed an atherogenic diet, a basic diet, or an atherogenic diet supplemented with an *Abies alba* trunk extract obtained by means of a two-step procedure (water at 70 °C followed by ethyl acetate), with an intake of about 10 mg/kg/day, equivalent to a human dose of about 150 mg/day. The addition of the *Abies alba* extract to the atherogenic diet significantly improved the aorta relaxation response compared to that of the atherogenic diet without the extract and significantly decreased the number of atherosclerotic plaques compared to the atherogenic group [77]. This dose translates into an effective proportion of *Abies alba* extract to DPI of 1:200 *w*/*w*. Lignans dominated the composition of extracts of *Abies alba* branches and twigs, with about 10% *w*/*w* [89], although with remarkable variability due to a steep content gradient from the proximal to distal sections [90].

Although further in vivo and clinical trials are urgently needed, the available evidence points to small relative quantities of *Abies alba* extracts needed to effectively functionalize both animal- and plant-based DPIs (1/150 to 1/200 *w*/*w*).

Table 3 summarizes the above figures for red orange waste peel, pomegranate waste peel, and *Abies alba* byproducts, with further information about the observed HC-based extraction yield.

The information presented in Table 3 about the amount of fresh raw resources is relevant for the assessment of the operating expenditure (OPEX), which directly affects the cost of goods sold (COGS) and, in turn, the price of the functionalized DPI. At the lowest effective dose (daily amount), the required amounts of fresh raw materials are relatively similar, with *Abies alba* byproducts compensating for the lowest extraction yield with the lowest moisture content.

Further information useful for assessing the COGS is the energy consumption required to obtain the dry bioactive extract. Based on data shown in a previous study [14], the specific energy consumption (energy consumed per unit mass of dry extract) depends on the moisture of the raw resource; the total extractables (i.e., the extraction yield in Table 3); the content of the dry biomass, i.e., the water-to-biomass ratio (dw); and the temperature range of the extraction process (shown in [14]). Assuming, for example, a fixed level of 10:1 for the water-to-biomass ratio (dw), the specific energy consumption would be 21.66, 17.53, and 46.34 kWh/kg of dry extract for red orange waste peel, pomegranate waste peel, and *Abies alba* twigs, respectively; the energy consumption required to produce the amounts to be added to a daily serving of DPI would be approximately 4.3 × 10^−3^ kWh (red orange waste peel), 4.4 × 10^−3^ to 19.3 × 10^−3^ kWh (pomegranate waste peel), and 7.0 × 10^−3^ to 9.3 × 10^−3^ kWh (*Abies alba* twigs). Based on the estimates of the amount of fresh raw resources and energy consumption required, on the economic side, the most favorable resource could be orange waste peel, followed by pomegranate waste peel and *Abies alba* twigs. However, the unit cost of the raw resources, which will be related to specific supply chains, and the consideration of specific target functionalities might lead to a change in the score.

An illustrative techno-economic note is presented in the Appendix A, based on the authors’ experience and confidential discussions with industry practitioners, for a representative dry extract obtained from the processing of red orange waste peel. The modeled COGS, representative of an industrial facility in Italy as of December 2025, includes all steps from the supply of raw materials to packaging and a warehouse with toll spray-drying and quality controls, as well as direct labor, depreciation, and maintenance of the equipment and overhead costs.

Appendix A shows the main parameters used in the calculations, excluding unit prices and other sensitive parameters due to their confidential nature; in particular, it was assumed that an annual quantity of dry extract of 200 kg would be produced, equivalent to 1 million effective daily doses, and that the equipment would be used for three concurrent production lines (e.g., red orange peel, pomegranate peel, and *Abies alba* byproducts).

Appendix A shows the impact of the water-to-biomass ratio (fw) on the total COGS per effective dose. Each unit increase in the ratio caused the COGS to increase by approximately EUR 0.008, with a nearly 60% increase from a ratio of 1 (EUR 0.041) to a ratio of 4 (EUR 0.065).

Appendix A show the share of the total COGS in reference to the fraction of the effective dose (daily amount) of dry extract and applicable to any amount of dry extract, corresponding to a water-to-biomass ratio (fw) of 4, 2, and 1. In agreement with a previous study [14], the cost contribution of energy and other utilities at the level of the local processing unit is only a tiny fraction of the COGS (less than 1%). Depreciation and maintenance dominate the share, followed by overheads, toll spray-drying, quality control and certifications, and direct labor. However, due to fewer processing cycles, the contribution of direct labor and quality control and certifications decrease with a decreasing water-to-biomass ratio (fw), and the contribution of toll spray-drying decreases due to the higher content of extractables per unit volume of water; thus, the relative contribution of depreciation and maintenance increases with a decreasing water-to-biomass ratio (fw).

The estimated sensitivity, for a given raw material, of both the total COGS and the relative share of the different cost elements to the biomass content per unit volume of water suggests two main lines of research as key to the possible future of HC-based technologies in the field of natural product extraction:Fundamental and industrial research should focus on efforts to manage processes with the highest possible biomass content, in particular overcoming the problem of viscosity both during extraction, as illustrated in [14], and in downstream steps (especially separation);Depreciation and maintenance represent a primary cost burden with a relative contribution to the COGS that increases with the biomass content, urgently requiring standardization as a path to substantial economies of scale, to be achieved through targeted experimental development and adoption in regular industrial processes, hopefully supported by forward-looking governmental or intergovernmental policies.

To enable these crucial developments, it is recommended that future studies report standardized process metrics (e.g., specific energy, temperature rise, solids loading, throughput) and, where possible, life cycle/techno-economic assessments.

Overall, while the estimated order of magnitude for the levels of COGS per effective dose of red orange peel dry extract of 200 mg (roughly, from EUR 0.040 to EUR 0.065) suggests the economic feasibility of the integration of HC-derived bioactive phytocomplexes in commercial DPIs, more consolidated data and an industrial history that has so far been insufficient are needed to explore the sensitivity of the COGS to process parameters and to derive reliable absolute values.

## 6. HC-Based Extraction of Vegetable Proteins

HC has rapidly moved from a proof of concept to pilot- and large-scale reality for extracting and upgrading plant proteins, offering shorter process times, lower solvent usage, and straightforward scalability compared with UAE, high-pressure processing (HPP), or purely thermal/alkaline routes. Beyond yield, HC can improve functional quality by mitigating anti-nutritional factors (ANFs) and preserving or enhancing protein structure, delivering ingredients that perform well in foods and nutraceuticals.

In legumes, comparative studies on pea demonstrated that both HC and UAE outperform conventional extraction in the recovery of protein isolates while better retaining structural integrity; however, HC is intrinsically scalable, whereas UAE faces geometric and energy-distribution limits at higher volumes, making HC the pragmatic choice for industrial throughputs [91]. Subsequent head-to-head work on ANF control showed HC to be the most effective technology for lowering trypsin inhibitor activity versus UAE and HPP, with phytic acid remaining the most persistent ANF—suggesting process windows that temper alkalinity or leverage near-neutral to mildly acidic media during or after HC to favor phytate solubilization or removal [25]. Together, these findings indicate that HC can deliver pea protein isolates that are not only high-yielding but also digestion-friendly and formulation-ready.

The value of HC extends beyond soft seeds to tough secondary streams. For fiber-rich oat hulls, two HC reactor designs achieved higher protein extraction than conventional methods using either dilute alkali or water alone while improving nutritional properties (higher in vitro digestibility and more favorable amino acid metrics). This suggests that the intense micro-mixing, shockwaves, and microjets generated by bubble collapse can compensate for limited solvent accessibility and reduce the need for harsh chemical treatments, as well as facilitate enzymatic digestion by reducing the particle size of the proteins [92]. The same processes, which are exclusive to HC on a full scale, were deemed responsible for the higher phenolic content and antioxidant activity both in the extracts and post digestion compared to conventional extraction.

Notably, both pea protein and oat hull isolates were obtained quite simply, adjusting the pH to the isoelectric point of the protein of 4.5 and separating by centrifugation, obtaining a protein content of about 80% and 56%, respectively [91,92].

Bench-to-pilot translation in other crops is consistent. For faba bean, integrating Osborne fractionation with pilot-scale HC boosted protein recovery compared to conventional extraction while simplifying the unit-operation train—an important lever for energy and OPEX reduction when moving beyond laboratory volumes [93]. In nuts, HC-based almond beverage processing showed that proteins can be liberated efficiently under water-based, short-residence conditions, with performance comparable to far more complex thermal/mechanical schemes; the authors noted that elevated temperatures and fast kinetics can cap ultimate recovery—pointing to the benefit of moderating the thermal load and extending the residence time, if allowed by the flavor and microbiological and lipid stability constraints [94]. Notably, it was found that the sensory properties of a whole-almond beverage, obtained by HC-based co-extraction of the almond kernel and its peel, were comparable to or even better than a high-end commercial product with the same water-to-almond ratio [14].

Apple processing offers a practical blueprint for circular integration. A recent study introduced the extraction of proteins from apple seeds for waste valorization [95], aligning naturally with HC pilot-scale work on apple pomace, where the same cavitation train already recovers pectin- and bioactives-rich streams [60]. Coprocessing seeds and pomace in a single HC workflow could therefore yield a defatted protein fraction from seeds, polyphenol- and pectin-rich aqueous extracts, and a fiber coproduct mainly consisting of cellulose—an attractive three-product cascade that maximizes raw-material value while keeping water as the dominant solvent.

Mechanistically, HC promotes rapid cell disruption and protein solubilization through repeated compression/rarefaction cycles and micro-scale shear, while the intense micro-mixing shortens diffusion paths and improves mass transfer. When mild alkali is used, HC accelerates unfolding and solubilization without prolonged exposure to high pH; when water-only processing is feasible, as shown for oat hulls, HC’s physical effects alone could be sufficient to achieve competitive yields with improved functional attributes [91,92]. Importantly, unlike acoustic systems, hydraulic reactors (e.g., static linear reactors, such as Venturi or orifice plates, or rotor–stator setups) scale linearly in flow and are compatible with inline heat exchange, pH control, and membrane clarification, enabling tight integration with downstream isolation and drying.

From a formulation standpoint, HC-processed plant proteins frequently exhibited better solubility and emulsification after ANF reduction and subtle structural rearrangements, along with higher phenolic content and antioxidant activity both in extracts and post digestion. This facilitated their combination with HC-generated carbohydrate and phenolic streams to create protein–phytocomplex ingredients with superior techno-functional and potential biological performance—a direction already emerging for soy and pea systems processed under cavitation [25,96].

In sum, the evidence across legumes, cereals, nuts, and fruit byproducts converges on a robust message: HC delivers high-quality vegetable proteins with fewer unit operations, lower chemical intensity, and credible industrial scalability. Specific priority optimizations for near-term deployment include the following:Tuning pH/ionic strength to tackle phytate without harming digestibility;Operating at moderate temperatures and solid loadings that preserve proteins while protecting flavor and lipids;Designing cascaded HC lines that co-valorize proteins, polysaccharides, and fibers from the same feedstock.

## 7. HC-Based Protein–Polyphenol Conjugation

The HC processing of citrus byproducts resulted in the stable conjugation of pectin and polyphenols, with the energy needed for the slightly endergonic reactions provided by the imploding cavitation bubbles [97], with the processes showing a high degree of reproducibility and the generated phytocomplexes exhibiting remarkable standardization and enhanced bioactivity, primarily due to the synergy between pectin and flavonoids and the increase in the bioavailability of flavonoids [8,98]. In the following, early evidence of the HC-driven conjugation of proteins and plant polyphenols is briefly reviewed, as it could positively affect the bioactivity of the resulting products.

### 7.1. Early Evidence of HC-Driven Protein–Polyphenol Complexation and Conjugation

HC promotes rapid unfolding of plant and dairy proteins and intense micro-mixing, enabling both non-covalent complexation (hydrogen bonding, hydrophobics) and covalent grafting to oxidized polyphenols (quinone-mediated Schiff base/Michael addition) within minutes. Recent HC studies with soy protein isolate (SPI) show formation of SPI–polyphenol complexes with enhanced structural order and interfacial activity, consistent with localized cavitation hotspots and shear-promoting exposure of reactive lysine and tyrosine side chains [96,99]. Cavitation microjets, shockwaves, and transient hotspots can promote covalent protein–polyphenol grafting via phenolic–quinone/amine (Schiff base and Michael addition) pathways and can accelerate radical-initiated conjugation, without bulk harshness.

As a mechanistic clarification, it is important to distinguish protein–polyphenol non-covalent complexation (electrostatic, hydrophobic, and π–π interactions and hydrogen bonding) from covalent conjugation, which requires formation of new covalent bonds and cannot be inferred from spectroscopic shifts alone. Reports describing cavitation-assisted conjugates often rely on Circular Dichroism (CD)/Fourier-Transform Infrared (FTIR)/fluorescence evidence consistent with unfolding and binding; these readouts are valuable but are not, by themselves, proof of covalent linkage [100]. Covalent conjugation is chemically plausible when polyphenols undergo oxidation (e.g., to quinones) followed by nucleophilic addition of protein side chains such as lysine, cysteine, and tyrosine (Lys/Cys/Tyr), but confirmation should rely on orthogonal evidence, ideally including MS-based identification of adducts, i.e., Liquid Chromatography–Tandem Mass Spectrometry (LC–MS/MS), alongside appropriate controls (pH/oxidation state, oxygen availability, scavengers) [101]. For this reason, we recommend reserving the term conjugation for cases supported by direct covalent bond evidence and using complexation/assembly when the data support binding-driven supramolecular structures without demonstrated covalent chemistry [100,101,102]. In practice, CD/FTIR/fluorescence support conformational change/binding, Sodium Dodecyl Sulfate–Polyacrylamide Gel Electrophoresis (SDS-PAGE) shifts support altered protein populations, and LC–MS/MS adduct mapping provides the most direct evidence of covalent conjugation.

Figure 2 shows the proposed HC-enabled pathway for protein–polyphenol complexation/conjugation and the resulting functionality, with recommended analytical readouts to distinguish complexation from conjugation.

Processability is credible: whey protein–pectin complexes have already been scaled from the lab to a continuous technical-scale line [103], indicating transferable unit operations for HC-assisted conjugates. Overall, HC shows the potential to uniquely couple fast protein exposure, phenolic oxidation, and mass transfer in water-only media, aligning with the green extraction logic already established for polysaccharide–polyphenol conjugates.

### 7.2. Added Functionality of Protein–Polyphenol Conjugates

Stable protein–polyphenol conjugates unlock techno-functional gains directly relevant to nutraceuticals and beverages: higher water solubility and shifted isoelectric behavior with better dispersion at a neutral pH, stronger interfacial films and emulsifying stability, and markedly improved oxidative protection in lipid and carotenoid systems. These outcomes are reported across whey and plant proteins—e.g., proanthocyanidin–WPI conjugates forming robust Pickering shells and protecting β-carotene [104], enhanced resveratrol loading with preserved functionality via grafting [105], and conjugates that maintain antioxidant activity through digestion [106,107]. Similar benefits track with numeric gains already mentioned in HC contexts for solubility, emulsification, and lipid-oxidation suppression in pea/soy/whey systems, reinforcing translational value for fortified foods and delivery systems. Together, these functionalities support cleaner labels (e.g., less surfactants/antioxidants), better bioaccessibility, and shelf-life extension—compelling complements to the HC polysaccharide-rich extracts highlighted in Section 3.7 and Section 4.

## 8. Conclusions

HC has advanced into a practical, water-centric unit operation that can deliver high-solids extraction and preserve functionality across diverse matrices, including citrus, pomegranate, and softwood byproducts. Two near-term, industry-ready routes are particularly compelling: (i) direct blending of HC-derived dry phytocomplexes with DPI products to enable low-dose fortification with measurable antioxidant, anti-inflammatory, or cardiometabolic effects and (ii) HC-based extraction of vegetable proteins that reduces anti-nutritional factors and yields isolates suitable for protein–polyphenol complexation, improving solubility, interfacial performance, and bioaccessibility. These routes leverage established separation and dry-mixing infrastructure, minimize added unit operations, and align with circular valorization of co-streams. While water management and drying remain important design levers, HC’s decisive advantage is coupling scalable, water-only processing with ingredient performance and straightforward integration, supporting broader adoption in nutraceuticals, foods, and related materials, when energy sources and accounting, quality assurance, and regulatory aspects are properly addressed.

This narrative review may be subject to selection bias despite systematic intent. Direct method-to-method comparison is limited by heterogeneity in feedstocks, reactor designs, operating conditions, and analytical endpoints. Reporting of energy metrics is often incomplete, and relatively few studies provide explicit solvent and byproduct mass balances, life cycle assessment, or techno-economic analyses. Mechanistic attribution of in situ conjugation is evolving, and evidence from continuous, multi-ton implementations is still sparse relative to pilot-scale reports.

Table 4 provides a concise Strengths/Weaknesses/Opportunities/Threats (SWOTs) synthesis for HC extraction at scale.

## Figures and Tables

**Figure 1 molecules-31-00192-f001:**
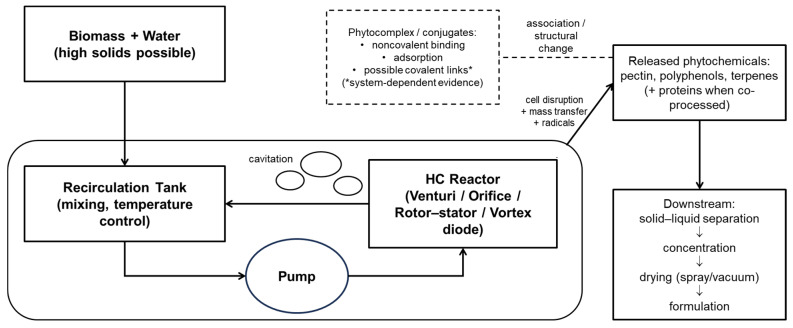
HC process flow for green extraction and phytocomplex formation. Typical reactor families include Venturi tubes, orifice plates, and rotor–stator/vortex devices.

**Figure 2 molecules-31-00192-f002:**
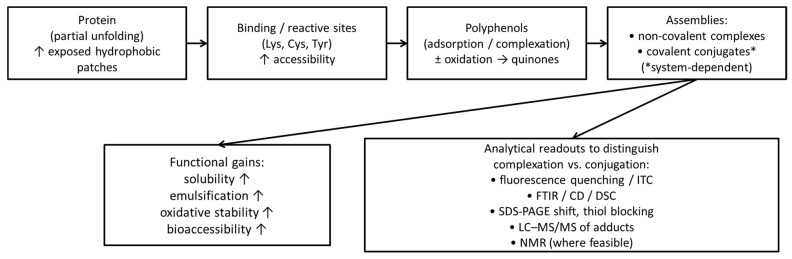
Proposed HC-enabled pathway for protein–polyphenol complexation/conjugation and resulting functionality, with recommended analytical readouts. ITC stands for Isothermal Titration Calorimetry; DSC stands for Differential Scanning Calorimetry; NMR stands for Nuclear Magnetic Resonance. The symbol ↑ beside certain properties stands for an increase in the same properties.

**Table 1 molecules-31-00192-t001:** Main advantages and drawbacks of CPW extraction methods.

Method	Advantages	Drawbacks
UAE	Water can be the only solvent; Low working temperature, fast, and low energy consumption.	Scaling beyond pilot is challenging due to acoustic field attenuation; Preservation of bioactive compounds sensitive to working temperature, amplitude, frequency, and power.
MAE	Low working temperature; Low energy consumption; High extraction yield; Preservation of bioactive compounds including volatiles.	Scalability not proven; Generally a pre-treatment and needs a further extraction technique downstream; High cost of equipment at the real scale.
PEF	Water as the only solvent; Very short processing time.	Generally a pre-treatment and needs a further extraction technique downstream.
SWE	Water as the only solvent; Selective extraction; continuous flow of operation; short time.	Difficult cleaning; Possible degradation of bioactive compounds due to high temperature and pressure; High cost of equipment; Energy intensive.
EAE	High quality of recovered pectin; As a pre-treatment, allows UAE to increase the extraction yield of phenolic compounds.	Lower recovery of phenolic compounds compared with conventional Soxhlet technique; Selectivity of enzymes; Long process time; Difficult to scale up; High cost of enzymes at the real scale.
NADESs	High selectivity of extracted bioactive compounds; Low working temperature; Simple equipment.	Scalability not proven; High cost of NADESs; NADES residues in the end product.
HC	Water as the only solvent; Low working temperature, fast, and low energy consumption; Creation of new stably conjugated, water-soluble phytocomplexes with higher bioavailability compared to individual compounds; Insoluble residues with high technical value; Straightforwardly scalable.	Non-standard equipment; Critical dependence of performance on construction details, hence the need for new skills.

**Table 2 molecules-31-00192-t002:** Assessment of the compliance of extraction technologies with the 12 GEPs (✓, likely compliant; ✗, likely non-compliant) [26].

GEPs	HC	UAE	MAE	PEF	SWE ^h^	EAE	NADES
1. Use water/safe solvents ^a^	✓	✓	✓	✓	✓	✗	✗
2. Non-denaturing conditions	✓	✓	✓	✓	✗	✓	✓
3. Minimize biomass pre-treatment	✓	✓	✓	✓	✓	✓	✓
4. Minimize energy consumption	✓	✓	✓	✓	✗	✓	✓
5. Renewable energy sources ^b^	✓	✓	✓	✓	✗	✓	✓
6. Minimize unit operations	✓	✓	✗	✗	✓	✓	✓
7. Integration with downstream ^c^	✓	✗	✗	✗	✗	✗	✗
8. Predictability and scalability	✓	✗	✗	✓	✓	✗	✗
9. Automation ^d^	✓	✓	✓	✓	✓	✗	✗
10. Safety and hygiene ^e^	✓	✓	✓	✓	✗	✗	✗
11. Valorize all byproducts ^f^	✓	✓	✗	✗	✗	✗	✗
12. Carbon footprint reduction ^g^	✓	✓	✓	✓	✗	✗	✗

^a^ Under conditions of the highest extraction rate. ^b^ Use of renewable energy sources, although enabled by the electric power source (except for SWE), depends on the facility and utilities. ^c^ Downstream integration favors HC due to more direct coupling to clarification and drying with minimal intermediate steps. ^d^ Automation is inherently context-dependent, still unfeasible for non-consolidated methods. ^e^ Safety and hygiene are hindered by high working temperature/pressure and use of enzymes and solvents needing post-extraction recovery. ^f^ Valorization of byproducts has been shown only for HC and UAE; the use of enzymes and solvents makes reuse of byproducts impractical. ^g^ Carbon footprint reduction is mainly determined by the carbon intensity of the utilities (electricity/heat) and by downstream operations (especially solvent recovery and drying); a quantitative comparison ultimately requires LCA/TEA and site-specific energy mixes. ^h^ SWE scores hinge on operating temperature/pressure and cleaning burden.

**Table 3 molecules-31-00192-t003:** Effective amounts of dry bioactive extracts per daily serving of DPI (30 g), reference molecule where available, HC-based yield of dry extract, and the necessary quantity of fresh raw material.

Raw Resource	Moisture ^a^ (%)	Daily Amount (mg)	Reference Molecule (Amount in mg)	Yield ^b^ (%)	Fresh Raw Material (g Wet Basis)
Red orange waste peel	75	200	Hesperidin (5)	30	2668
Pomegranate waste peel	72	250–1100	Punicalagin (35–75)	35 ^c^	2551–11.224
*Abies alba* byproducts	30 ^d^	150–200	Lignans (9–24) ^e^	11 ^d^	1949–2597

^a^ Moisture of raw resource, from [14]. ^b^ HC-based extraction yield of dry raw resource, from [14]. ^c^ Updated from [14], based on experiments performed with the upgraded device referred to as HC300 in [60]. ^d^ Refers to branches/twigs, lignans dominated by secoisolariciresinol and isolariciresinol, and accounting for the steep gradient from proximal to distal sections (content assumed in the range 6–12% *w/w*). ^e^ Data available for *Abies alba* twigs.

**Table 4 molecules-31-00192-t004:** SWOTs synthesis for HC extraction at scale.

SWOTs Item	Note
Strengths	Water-centric processing (often no organic solvents); fast mass transfer and micro-mixing; compatibility with continuous recirculation loops; potential for integrated co-valorization (fibers, proteins, polysaccharides).
Weaknesses	Reactor designs and reporting remain heterogeneous; solids-rich slurries can challenge downstream separation; drying/finishing steps can dominate energy and COGS; erosion/material wear and hygiene/CIP constraints.
Opportunities	Integration in agro-industrial biorefineries; demand for clean-label extracts and functional ingredients; protein functionalization as an added-value route; standardization enabling TEA/LCA and wider adoption.
Threats	Over-claim risk without head-to-head datasets; scale-up without performance metrics can mislead; regulatory uncertainty for novel ingredients; feedstock seasonality/logistics and market (sensory) acceptance.

## Data Availability

Data are available within the article and its Appendix A.

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
