# Peer review of "Green Extraction at Scale: Hydrodynamic Cavitation for Bioactive Recovery and Protein Functionalization—A Narrative Review"

_molecules, 2026, doi:10.3390/molecules31010192_

Round 1
Reviewer 1 Report
Comments and Suggestions for Authors
The paper of Meneguzzo et al. “Green Extraction at Scale…” aimed to summarizing the information on the hydrodynamic cavitation application for some plant metabolite extraction.
Highlights and strengths of the manuscript are:
The results may further increase interest in the study of hydrodynamic cavitation in isolation / extraction of plant-based products / material in safe green conditions and use some of the findings to predict future research in this area of science.
Specific comments and suggested revisions:
- The authors attempted to compare the efficiency of HC with other methods used for green extraction. And although I understand the arbitrary nature of such a comparison, the authors should have mentioned that their conclusions are preliminary. The most reliable data are obtained by directly comparing the extraction efficiency of methods in one experiment for one object at one time. All the advantages and disadvantages indicated by the authors are not correct if the methods were not compared directly by one group of scientists. It is always possible to adjust the existing results to the desired concept.
- I will give a simple example from your table 2. The first line indicates the possibility of using water and safe solvents. For enzyme and NADES extraction, the probable impossibility of meeting this requirement is indicated. Why? For these methods, it is possible to use safe solvents. What does this lead to at the end of the table? This leads to a decrease in the level of compliance of the method, although this is not the case. Similar claims can be made for all of the 11 remaining points. I repeat that I understand the authors' desire to demonstrate the advantages of the HC, but the way of comparison they have chosen leads to false conclusions about the advantages of some methods and the disadvantages of others. I'm not even mentioning the cost of the equipment and the problems, for example, at the simplest stages of filtration (since a low hydromodulus leads to the formation of a viscous mixture, which hinders the separation of the liquid extract from the pulp).
- It's worth noting that the authors don't provide references to specific articles demonstrating the effectiveness of HC compared to others methods. This means your conclusions are simply speculation or a simulation.
- The choice of individual examples demonstrating the advantages of HC is somewhat surprising - orange peel, pomegranate peel, and silver fir (Abies alba) byproducts. Are there the largest number of articles on these cases, or is this topic closest to the authors?
- Let me clarify right away: this doesn't mean the article is invalid. It simply needs to address some contentious points, include proper references, and clarify that the comparison represents only preliminary conclusions about potential advantages of HC.
With all due respect to the authors, the article in its current form is quite controversial, but could be improved after correction.
Reviewer 2 Report
Comments and Suggestions for Authors
The review article submitted for review is interesting. The authors have well-designed its individual sections, beginning with a brief presentation of methods for extracting bioactive compounds classified as ecological, and then describing the Hydrodynamic Cavitation (HC) method, which is the subject of this review. The authors included orange peel, pomegranate peel, and silver fir byproducts as plant materials. They pointed out compounds that can be effectively extracted using Hydrodynamic Cavitation. Information was provided regarding the impact of this method on the potential for creating protein isolate complexes with polyphenols and the extraction of plant proteins using HC. I have only one comment: the authors used the term "natural products" five times. Only in one case did they clarify which products they were referring to. I request that this information be included in the remaining cases as well.
I rated the manuscript submitted for review exceptionally highly, as it is a review article that does not require any special methodology, unless it were based on a meta-analysis (this element was not included in this review). The authors briefly described the methods used to extract bioactive compounds, which are classified as green chemistry methods. However, they devoted more space to another less-known technique (Hydrodynamic Cavitation), which may be less relevant from the point of view of laboratory-scale application but is important for future macroscale applications. Both the tables presented and the references provided raise no objections. Similarly, the abstract and conclusions have been presented correctly.
Reviewer 3 Report
Comments and Suggestions for Authors
Bibliometrics is strongly recommended for analyzing the current research status.
When analyzing literatures, it is recommended to provide the search time range, database, keywords, etc.
At lease one figure (e.g., about the protein–polyphenol conjugates, HC process flow, reactor structure diagram or extraction path diagram) should be added for the review, because it can provide readers more clear introduction.
The mechanism of how HC promotes the formation of conjugates is not sufficiently elaborated.
The article mentions technical standardization multiple times, but does not delve into how to promote standardization in depth.
Although raw material costs and energy consumption, a complete lifecycle assessment or technical economic analysis were not conducted, resulting in an incomplete evaluation of greenness.
When compared HC with other technologies such as UAE, PEF, SWE, etc., there are more descriptions of its advantages, but insufficient discussions on the potential advantages of other technologies or the limitations of HC.
Due to the significant impact of different reactor/extractor designs on operating conditions such as pressure, temperature, and flow rate, could the author discuss this further?
SWOT mode is suggested for the Conclusions part.
Round 2
Reviewer 1 Report
Comments and Suggestions for Authors
The authors taking into account the most observations made by the reviewer. The reviewer still has doubts about whether all the stated goals have been achieved, but overall, the work appears to be coherent. If the editor thinks the work is suitable for publication, then I will not spoil everyone's mood and will agree with his decision. The only thing that bothers me is the high percentage of iThenticate. I believe that this deficiency should be corrected by the authors, after which the manuscript can be recommended for publication.
Reviewer 2 Report
Comments and Suggestions for Authors
I have no additional comments. I accept the manuscript in its current form.
Reviewer 3 Report
Comments and Suggestions for Authors
The abbreviations in supplemented contents have not been added in final list of abbreviations (e.g., those in Fig.2);
The symbols (e.g., *) in Figs.1-2 should be elucidated in its caption;
For the journal names of references, there are now both abbreviations and full names.
